# Automatic Segmentation of the Retinal Nerve Fiber Layer by Means of Mathematical Morphology and Deformable Models in 2D Optical Coherence Tomography Imaging

**DOI:** 10.3390/s21238027

**Published:** 2021-12-01

**Authors:** Rafael Berenguer-Vidal, Rafael Verdú-Monedero, Juan Morales-Sánchez, Inmaculada Sellés-Navarro, Rocío del Amor, Gabriel García, Valery Naranjo

**Affiliations:** 1Departamento de Ciencias Politécnicas, Campus de los Jerónimos, Universidad Católica de Murcia UCAM, 30107 Murcia, Spain; rberenguer@ucam.edu; 2Departamento de Tecnologías de la Información y Comunicaciones, Campus Muralla del Mar, Universidad Politécnica de Cartagena, 30202 Cartagena, Spain; juan.morales@upct.es; 3Servicio de Oftalmología, Hospital General Universitario Reina Sofía, 30003 Murcia, Spain; inmasell@um.es; 4Instituto de Investigación e Innovación en Bioingeniería, Universitat Politècnica de València, 46022 Valencia, Spain; madeam2@upvnet.upv.es (R.d.A.); jogarpa7@i3b.upv.es (G.G.); vnaranjo@dcom.upv.es (V.N.)

**Keywords:** optical coherence tomography (OCT), peripapillary OCT, automatic layer segmentation, retinal imaging analysis, mathematical morphology, active contours, glaucoma

## Abstract

Glaucoma is a neurodegenerative disease process that leads to progressive damage of the optic nerve to produce visual impairment and blindness. Spectral-domain OCT technology enables peripapillary circular scans of the retina and the measurement of the thickness of the retinal nerve fiber layer (RNFL) for the assessment of the disease status or progression in glaucoma patients. This paper describes a new approach to segment and measure the retinal nerve fiber layer in peripapillary OCT images. The proposed method consists of two stages. In the first one, morphological operators robustly detect the coarse location of the layer boundaries, despite the speckle noise and diverse artifacts in the OCT image. In the second stage, deformable models are initialized with the results of the previous stage to perform a fine segmentation of the boundaries, providing an accurate measurement of the entire RNFL. The results of the RNFL segmentation were qualitatively assessed by ophthalmologists, and the measurements of the thickness of the RNFL were quantitatively compared with those provided by the OCT inbuilt software as well as the state-of-the-art methods.

## 1. Introduction

Glaucoma is one of the leading causes of irreversible blindness worldwide [1]. The global prevalence of glaucoma for the population aged 40–80 years is 3.54%. In 2013, the number of people (aged 40–80 years) with glaucoma worldwide was estimated to be 64.3 million, increasing to 76.0 million in 2020 and 111.8 million in 2040 [2]. This silent ocular disease of the optic nerve head is usually caused by a high intraocular pressure due to poor drainage of ocular fluid [3], producing a progressive and irreversible deterioration of the visual field that progresses to a total loss of vision [4].

Optical coherence tomography (OCT) has become an important tool in ophthalmology for eye diagnostics, especially for the anterior segment or the retina [5]. The OCT is based on the estimation of the depth at which a specific backscatter is originated by measuring its time of flight through interferometry [6]. These backscatters are caused by differences in the refractive index between adjacent tissues and enable 2D- and 3D-imaging with resolution on the micrometer scale [7]. Spectral-domain OCT technology provides 2D and 3D scans of the retina allowing for monitoring eye diseases such as glaucoma [8], since it allows for analyzing and quantifying regions such as the retinal nerve fiber layer (RNFL) and ganglion cell inner plexiform layer (GCIPL). Among the retinal OCT imaging modalities, there are 3D or volumetric OCTs (usually centered on the macula); 2D peripapillary (or circumpapillary) B-scan OCTs centered on the optic disc; and 2D radial (non peripapillary) B-scan OCT crossing the macula or crossing the optic disc (or optic nerve head). To assess the glaucomatous status or progression, the OCT of the papillary circle (2D peripapillary B-scan OCT centered at the optic nerve head) allows for the determination of the thickness of the peripapillary nerve fiber layer. The thickness of the RNFL measured in peripapillary circular OCT scans is an indirect way of assessing the generalized damage that glaucoma produces in the retinal ganglion cells [9], since the narrowing of the RNFL precedes visual field loss and optic nerve head defects [10,11]. Some of the most popular (and publicly available) databases that can be found are, e.g., Duke SD-OCT [12,13], with 110 OCT B-scans centered at the macula; the dataset described in [14,15], which collects 12 radial OCT B-scans for 61 different subjects; the annotated retinal OCT images (AROI) database [16,17], which is composed of OCT volumes centered on the macula to study the neovascular age-related macular degeneration (nAMD); the OCTID database [18,19], containing retinal fovea-centered 2D OCT images with different pathological conditions; and the dataset contained in [20,21], with volumetric OCTs centered on the macula of left and right eyes of 50 healthy persons. Most of those references are devoted to providing the thickness map of retinal layers, including the RNFL. However, none of the above works contain 2D peripapillary B-scan OCTs, imaging modality used in the proposed approach.

In recent years, the segmentation and analysis of retinal layers in OCT images and their clinical implications have focused the efforts of many researchers [22]. Besides the RNFL, researchers particularize their methods to detect and segment in OCT images some retinal abnormalities such as exudate-associated derangements [23], drusen volume [24,25], neovascularizations [26], intraretinal cysts [27], or to assess the presence and quantity of intraretinal and subretinal fluids [28] and image-guided treatment management [29,30]. Focusing on the RFNL, it was initially differentiated from other retinal layers in OCT imaging using a thresholding algorithm that detected the anterior edge of retinal pigment epithelium and determined the photoreceptor layer position. For each column of the image, the posterior edge was determined to be the first occurrence of the signal above the threshold anterior to the RNFL. Then, in the OCT image, the nerve fiber layer thickness at each column was defined as the number of pixels between the anterior and posterior edges of the RNFL. To achieve this objective, different approaches were followed to segment the RNFL, some of them providing methods that allowed (or required) the interaction with the expert, and other approaches that turn out to be automatic methods.

Regarding manual and interactive methods, the usual procedure to delineate the fiber layer or different lesions in the OCT images is to apply an initial preprocessing and then, after manual or interactive initialization, to carry out an automatic process. For example, in [31], this scheme is followed using anisotropic diffusion to reduce the speckle noise while at the same time preserving the edges and deformable models with a manual initialization to calculate the retinal thickness and delineate fluid-filled regions. Following the same plan, in [32], the semiautomatic segmentation of the retinal layers is performed from the contours interactively specified and refined by a clinician using a support vector machine (SVM) to handle OCT noise, considering statistical characteristics at multiple levels of resolution.

The quantitative evaluation of representative features of the retina with a fully automatic segmentation can be addressed from many perspectives with very different methods, as those based on, e.g., dynamic programming [33], Markov models [34,35], deformable models [36,37], graph search algorithm [38,39], level sets [35,40], fuzzy histogram hyperbolization [41] or geodesic distance [42]. Approaches based on machine learning were also used to automatically segment and analyze the retinal layers in OCT images. For example, a surrogate-assisted method based on the convolutional neural networks was proposed in [43] to classify retinal OCT images; in [44], tailored prototypical neural networks were developed for glaucoma grading using raw circumpapillary B-scans; the effective features of the boundaries were extracted in [45] with a convolutional neural network to obtain the final retinal boundaries using a graph-based search on the probability maps; in [46], the characteristics of retinal layers were learned with a deep neural network in a multiscale approach performing the segmentation with an encoder-decoder module; layers and fluid in 3-D OCT retinal images of subjects suffering from central serous retinopathy were simultaneously segmented in [47] with random forest classifiers; the most significant retinal layers in rodent OCT images were detected in [48] with a encoder-decoder fully convolutional network (FCN) architecture. Recently, in [49], the order of the layers was included explicitly in their deep learning method, since most of previous approaches did not consider it and this could lead to topological errors.

After this introduction, the following sections of this paper describe the proposed novel approach. The first part of the method, detailed in Section 2.2, is based on mathematical morphology, with it being used to clean the OCT image and robustly detect the RNFL boundaries; the second part of the method is described in Section 2.3 and details the implementation of deformable models to accurately delineate the contours and then provide the measurements of the RNFL thickness. Section 3 contains the results provided by this method, which have been qualitatively supervised and validated by ophthalmologists and quantitatively compared with those provided by the OCT device as well as other state-of-the-art methods based on deep learning. The upper boundary of the RNFL is perfectly delineated by the proposed method, similar to the segmentation of the OCT inbuild software, while the method based on deep learning generates some contours with certain disparities due to speckle noise. The lower contour provided by the three methods does differ due to artifacts and shadows produced by blood vessels, and each method must impose the constraints of its model. The proposed method incorporates the elasticity and stiffness characteristics of the RNFL in the active contour to segment the boundaries of this retinal layer. As a relevant contribution of this work and as an important aspect that makes the proposed method remarkable is, apart from the quality and precision of the segmentations, the computational efficiency, providing robust results in significantly less time than the method based on deep learning. This efficiency is achieved thanks to the optimal implementation of the morphological filters [50] as well as the formulation and implementation of the deformable models that are carried out in the frequency domain [51,52], using the dynamic parameters able to ensure and optimize the convergence of the model [53,54]. Finally, Section 4 proceeds with the discussion of the results, and Section 5 closes the paper with the conclusions and proposed future work.

## 2. Materials and Methods

This section details the materials used in this work, i.e., 2D peripapillary B-scan optical coherence tomography (OCT) images centered on the optic disc, along with their acquisition procedure, as well as the proposed method for the segmentation of the retinal nerve fiber layer (RNFL) in these OCT images, which relies mainly on mathematical morphology and deformable models. The procedure for the calculation of the layer thickness can be briefly outlined in the following blocks:Preprocessing to reduce the speckle noise and undesired artifacts in peripapillary 2D B-scan OCT images. This filtering is mainly performed by morphological operators.Peak detection on positive and negative part of the vertical gradient. This column-wise process (1D scenario) provides a rough detection of the layer’s boundaries.Reduction of possible discontinuities in previous coarse detection by processing adjacent columns (2D scenario) with a sliding window low-pass filter.Refinement of previous result by the imposition of the physical characteristics of biological tissues. The properties of elasticity and rigidity of the solution are incorporated by using a deformable model (in a 2D scenario).

The novel approach proposed in this paper provides an efficient and robust method for measuring the thickness of the RNFL on peripapillary 2D B-scan OCT images. The efficiency of the method is achieved thanks to the frequency implementation of the active contours and the optimized implementation of the morphological operators.

### 2.1. Materials: OCT Database

The main imaging modality used in the diagnosis and evolution of glaucoma is the OCT of the papillary circle (2D peripapillary B-scan OCT centered at the optic nerve head), which allows for the determination of the thickness of the peripapillary nerve fiber layer. For that reason, a set of spectral domain-OCT (SD-OCT) image pairs of both the left (162 images) and right eye (167 images) including healthy and unhealthy patients (glaucomatous optic neuropathy) was collected in this work. The images were acquired by the Ophthalmology Service of the Hospital General Universitario Reina Sofía (Murcia, Spain) using a Spectralis OCT S2610-CB (Heidelberg Engineering GmbH, Heidelberg, Germany) from October 2018 to November 2020. All images were anonymized in order to fulfill the criteria of the Human Ethics Committee. Table 1 details the content of the image set. This dataset is a batch of the database used to validate the method. The dataset will be completed and compiled, and it will be published in a publicly accessible database in the future.

The scanning procedure is based on spectral domain optical coherence tomography (SD-OCT) technology, also referred to as Fourier domain OCT (FD-OCT). It uses a beam of a superluminescence diode (SLD) with an average wavelength of 870 nm to take high-resolution cross-sectional imaging of the retina, also known as a B-scan image. This B-scan allows the thickness of the distinctive layers of the retina to be determined. These layers can be observed as a set of fairly flat bands in different grayscale values. Although different scanning patterns are allowed, to assess the glaucoma disease, a circular shape centered on the optic disc is usually used in the B-scan OCT section, as shown in Figure 1.

Note that this cylindrical section is projected from polar to Cartesian coordinates for easier processing and analysis. This projection together with its corresponding *x*- and *z*-scaling is displayed in the top right of Figure 2. The correspondence between the sectors of the analysis circumference and the *x*-coordinate of the peripapillary OCT images is gathered in Table 2 and illustrated in Figure 3.

The OCT images used in this work have a resolution of 768 × 496 pixels with a *z*-scaling of 3.87 µm/pixel and a bit depth of 8 bits/pixel in grayscale. The diameter of the peripapillary OCT circumference can be adjusted according to the size of the patient’s eye, varying the horizontal resolution of the OCT accordingly. For instance, a circumference diameter of 3.7 mm corresponds to a *x*-scaling of 15.21 µm/pixel. Note that only the *z*-scaling was used for this work, since the main goal is to determine the vertical thickness of the RNFL in the OCT image. The top left of Figure 2 shows the circular tracing of the SLD beam on the retinal fundus image of the eye. This analysis circumference is positioned over the center of the optic nerve (or papilla) as depicted in the figure.

In order to obtain the thickness of the RNFL layer, it is first necessary to accurately segment it. The automatic segmentation of this layer can be very challenging due to the characteristics of the image in general and of the RNFL in particular. As depicted in Figure 3, OCT images often show a speckle noise, a low level of contrast, and irregularly shaped morphological features [5,6].

As can be seen in Figure 3 and Figure 4, the RNFL is the innermost layer of the retina. Granular noise and various artifacts appear in the area of the image above the RNFL, which corresponds to the vitreous body. These artifacts have similar grayscale levels to the layer of interest, which leads to a more challenging segmentation process. Moreover, the transition between the RNFL and the CGL, which is located just below the RNFL, shows a low level of contrast, making the separation between these two layers a relevant issue [22].

In addition, because blood vessels are located just above the retina, a shadowing effect appears on the retinal layers. As observed in Figure 5, these vessels cross the circumference of analysis perpendicularly, casting a shadow on the retinal layers at these crossing points. This results in intensity inhomogeneities, boundary artifacts, and darkened areas in the layers to be segmented, as portrayed in Figure 6. This further complicates the segmentation of the RNFL and hence the calculation of its thickness.

### 2.2. Preprocessing: Mathematical Morphology

In order to minimize the effect of these image artifacts on subsequent segmentation, a mathematical morphology-based preprocessing was first applied to these images. Mathematical morphology is a nonlinear image processing methodology whose formulation and implementation is focused on the shape of the objects in the image [55,56]. The application of morphological filtering to an image consists of the use of so-called morphological operators. These operators imply the application of a structuring element, which is usually designed as a simple binary image with a specific shape and size. Depending on both the structuring element and the morphological operator applied, very different results can be obtained [50]. The main strength of mathematical morphology is the ability to minimize artifacts and uninteresting elements while maintaining the shape as well as the boundaries of the objects of interest.

Four different shapes are used in this work as structuring elements, namely a circle of radius *r* pixels, ${◯}_{r}$, a rectangle of width *a* pixels and height *b* pixels, ${□}_{a\times b}$, and horizontal $l_{c}^{h}$ and vertical $l_{c}^{v}$ lines, where *c* denotes their length in pixels.

Regarding the morphological operators, opening, closing, dilation, and reconstruction are used in this work. The opening $\gamma_{⋄}$, where ⋄ represents the structuring element used, allows bright details smaller than the structuring element to be removed from the image. Similarly, by applying the closing $\varphi_{⋄}$, dark elements smaller than ⋄ can be removed from the data. The application of the dilation operation, $\delta_{⋄}$, to a binary image allows the non-zero areas of the image to be expanded by the size and shape of the structuring element ⋄. The morphological reconstruction $R_{g}\left(f\right)$ restores the original shapes of the objects of the marker *f*, constrained by the mask *g*. For a more detailed description of these operators and their potential applications, see [50]. Additionally, the image complement $f^{N}$ is also used in this preprocessing step, which consists of the negative of an image *f* using unsigned integer format, i.e., $f^{N}=255-f$.

### 2.3. Boundary Segmentation: Active Contours

After preprocessing the image, minimizing artifacts and blood vessel shadows, and highlighting the layer of interest to be analyzed, the RNFL was segmented. To this end, the upper and lower boundaries delimiting the layer must be outlined. It is important to note that since the RNFL is a human tissue, the main features of these boundaries are their smoothness and continuity. For this reason, the application of deformable models to the preprocessed images from the previous stage is of considerable interest.

Deformable models are mathematical curves or surfaces whose shapes is conditioned by internal forces related to the shape of the object itself and external forces from the data set [57]. The internal forces are calculated from the geometry of the model and tend to preserve its smoothness. The external forces allow the model to be adjusted to the boundaries of the objects of interest in the image. This is the main reason why these models have been used for years in segmentation and the tasks of structure tracking in images [58,59,60].

Deformable models can be classified into explicit and implicit ones. In the explicit models [61,62], also known as parametric deformable models, the object to be segmented is represented by meshes (curves, surfaces, or volumes) whose shape evolves under the control of these internal and external forces. By contrast, in the implicit models [63,64], also called geometric deformable models, contours and surfaces are implicitly represented as level-sets of a scalar function of higher dimensions. The evolution of these implicit models is based on the theory of curve evolution, which allows their adaptation even to concave shapes and topology changes [63,65]. Since the retinal layers to be segmented in this work show a smooth shape and no change in topology, one-dimensional parametric models, also called active contours or snakes [61], are used in this work.

Active contours are described by a parametric curve,
(1)$$v\left(s,t\right)=[v_{x}\left(s,t\right),v_{y}\left(s,t\right)],$$
defined in $R^{2}$, where *t* denotes the time, *s* is the parametric variable with s∈[0,S], and $v_{i}\left(s,t\right)$ represents the coordinate function for coordinate *i*.

The shape and position of the curve v(s,t) was determined from the minimization of an energy functional defined by a second-order partial differential equation system, which incorporates the abovementioned internal and external forces [59]. The internal ones emulate the physical characteristics of the objects in the image, such as the elasticity or stiffness. In this work, the parameters of the deformable model were set so that the contour had topological features similar to those expected in retinal layers, in particular, its smoothness. External forces were used instead to fit the contour to the layer boundary, based on the information provided by the OCT imaging. These external forces were calculated by applying a vertical gradient to the image preprocessed with the morphological operators described in Section 2.2.

An efficient frequency-domain-based implementation [51,52] was used in the proposed approach. To derive this formulation, a finite element discretization was firstly applied to each coordinate function of the curve,
(2)$$v_{i}\left(s,t\right)=N\left(s\right)u_{i}\left(t\right),$$
where N(s) is the shape function and $u_{i}\left(t\right)$ are the nodal variables of the coordinate *i*. B-splines are used here as a shape function, due to their controlled smoothness of the resulting active contours [66]. Note that for this work, the *x*-coordinate of the nodal variables is fixed, and the only variation on the *y*-coordinate of the node position is allowed. Therefore, the deformable model is applied only to the *y*-coordinate, and, for the sake of clarity, the subscript *i* is omitted in the following expressions.

Subsequently, time is discretized t=ξΔt, where Δt is the time step and ξ∈N the iteration index. The parametric variable *s* is then translated into the frequency domain, resulting in a iterative equation which rules the adaptation of the curve to the dataset,
(3)$$U_{\xi }\left(\omega \right)=H\left(\omega \right)\left(a_{1}U_{\xi -1}\left(\omega \right)+a_{2}\;U_{\xi -2}\left(\omega \right)+\frac{Q_{\xi -1}\left(\omega \right)}{\eta F(\omega )}\right),$$
where ω is the frequency counterpart of the parametric variable *s* and ξ is the iteration index. The terms $U_{\xi }\left(\omega \right)$ and F(ω) are the Fourier transforms of nodal variables u and the shape function of the spatial discretization, respectively. Internal forces due to physical features of the curve are imposed by the filter H(ω), which is calculated from the elasticity and stiffness parameters of the model, α and β, respectively. The external forces are computed from the gradient of the preprocessed OCT image and gathered in the term $Q_{\xi -1}\left(\omega \right)$. Finally, the first and second order coefficients of the iterative system $a_{1}$ and $a_{2}$, depend on the parameters η and γ. These parameters control the inertia of the model in the process of adaptation to the data. The optimal values of all these parameters to accelerate the convergence process of the active contour can be found in [53,54].

### 2.4. RNFL Layer Segmentation Process

The proposed approach for the segmentation of the RNFL consists of a sequential process for obtaining different layer boundaries. Figure 4 illustrates the three boundaries used in the segmentation of the RNFL layer, namely #1-UB, #2-AB, and #3-LB.

The upper boundary of the RNFL, #1-UB, was outlined first, taking advantage of the high-contrast level between this layer and the upper area of the image. However, morphological preprocessing and the subsequent approximation with active contours are necessary in order to minimize the artifacts present in a large number of OCT images of the database. Section 2.5 describes this process in detail.

Obtaining the lower boundary of the RNFL, #3-LB involves a more challenging process, due to shadowing and artifacts caused by blood vessels as well as the low level of contrast between the RNFL and the lower layer, GCL. For this reason, the lower boundary of the ONL, #2-AB, was outlined and used as an auxiliary boundary which together with #1-UB, allowed us to obtain #3-LB in a third stage. Section 2.6 and Section 2.7 detail the estimation process of #2-AB and #3-LB, respectively, in which the morphological preprocessing and active contour smoothing were used again.

### 2.5. Segmentation of Boundary #1-UB

To outline #1-UB, a morphological processing was first conducted using a circle of radius 7 as a structuring element. Figure 7 illustrates this procedure. An opening was applied to the OCT in order to suppress bright details smaller than ${◯}_{7}$,
(4)$$I_{1}^{\#1}=\gamma_{{◯}_{7}}\left(I\right),$$
where *I* stands for the OCT image to be segmented. As can be seen, the small bright elements of Figure 7a are suppressed in Figure 7b. In addition, most of the speckle noise above the RNFL was significantly reduced.

A vertical gradient was then applied to the image, removing all negative values,
(5)$$I_{2}^{\#1}=\nabla_{v}^{+}\left(I_{1}^{\#1}\right)=\left\{\begin{array}{cc} \nabla_{v}\left(I_{1}^{\#1}\right), & 0\leq \nabla_{v}\left(I_{1}^{\#1}\right),\\ 0, & \nabla_{v}\left(I_{1}^{\#1}\right)<0, \end{array}\right.$$
where $\nabla_{v}\left(I_{1}^{\#1}\right)$ represents the gradient of the image $I_{1}^{\#1}$ along the vertical direction. This allows one to detect the black-to-white transitions in the vertical downward direction as shown in Figure 7c.

We parsed the gradient values in columns in the downward direction, selecting the first value in the column exceeding a specific threshold, which provides the boundary shown in Figure 7d. As can be seen, the estimated contour roughly followed the upper boundary of the RNFL. Nevertheless, some artifacts were visible at some points of the boundary due to remaining granular noise in the area of the image above the layer.

In order to minimize these artifacts, an active contour approach was then applied. We first smoothed the detected points by applying a low-pass filtering with a sliding window with width N=101 pixels [67]. The smoothed contour, which is shown in Figure 7e, was used as the initialization of the deformable model. An active contour was then applied using the parameters α=0.5, β=1, γ=1, η=1, B-splines as shape function, and the image $I_{1}^{\#1}$ as source for the external forces calculation. Figure 7f depicts the resulting contour, which fits the upper boundary of the RNFL.

### 2.6. Segmentation of Boundary #2-AB

For the delineation of #2-AB, we first applied a set of morphological operators using two circles with radii of 3 and 9 pixels as structure elements. Figure 8 illustrates the operations performed on the OCT image.

Dark details smaller than ${◯}_{9}$ were eliminated first. This implies the application of the following morphological operators. Initially, we took the complement of the OCT image *I*,
(6)$$I_{1}^{\#2}=f_{N}\left(I\right).$$

Next, an opening with a ${◯}_{9}$ was applied,
(7)$$I_{2}^{\#2}=\gamma_{{◯}_{9}}\left(I_{1}^{\#2}\right),$$
followed by a image reconstruction of the marker $I_{2}^{\#2}$ with the mask $I_{1}^{\#2}$,
(8)$$I_{3}^{\#2}=R_{I_{1}^{\#2}}\left(I_{2}^{\#2}\right),$$
and a new image complement,
(9)$$I_{4}^{\#2}=f_{N}\left(I_{3}^{\#2}\right).$$

As can be clearly seen in Figure 8e, the dark details smaller than the structuring element were eliminated while maintaining the rest of the image areas. The next action was to remove the bright details also smaller than ${◯}_{9}$. For this purpose, an opening was applied,
(10)$$I_{5}^{\#2}=\gamma_{{◯}_{9}}\left(I_{4}^{\#2}\right),$$
followed by an image reconstruction of the marker $I_{5}^{\#2}$ with the mask $I_{4}^{\#2}$,
(11)$$I_{6}^{\#2}=R_{I_{4}^{\#2}}\left(I_{5}^{\#2}\right).$$

Figure 8g shows the result of the application of these operators. Finally, we further refined the image by removing details, both light and dark, smaller than ${◯}_{3}$, by applying an opening,
(12)$$I_{7}^{\#2}=\gamma_{{◯}_{3}}\left(I_{6}^{\#2}\right),$$
and a closing,
(13)$$I_{8}^{\#2}=\varphi_{{◯}_{3}}\left(I_{7}^{\#2}\right),$$
which is depicted in Figure 8h,i. As can be observed, the application of these morphological operators led to the removal of all artifacts from the ONL layer, and a clear definition of the lower margin of the ONL layer, which is used as the boundary #2-AB. In order to contour this boundary, the positive vertical gradient defined in Equation (Equation 5) was calculated,
(14)$$I_{9}^{\#2}=\nabla_{v}^{+}\left(I_{8}^{\#2}\right),$$
which is shown in Figure 8j. Since this operation allowed us to obtain all black-to-white transitions in the vertical downward direction, several edges appeared in the image. It was therefore necessary to select the boundary of interest. This process was performed again column-wise. Figure 9 plots the value of one such column of pixels. The *x*-axis represents the vertical position, where values 1 and 496 represent the top and bottom row of pixels in the image, respectively. The *y*-axis shows the intensity level of the pixels. Since the OCTs of this work have a bit depth of 8 bits/pixel, the pixel values are in the range [0, 255], with the minimum and maximum values corresponding to the black and white tones, respectively. Note that each of the edges depicted in Figure 8j appears in Figure 9 as a local maximum.

We defined the intervals $M_{x}=\left[\varrho +\rho_{1},\varrho +\rho_{2}\right]$, where ϱ is the vertical position of #1-UB in that column, and $M_{y}=\left[\gamma ,255\right]$, where γ is a threshold determined from the empirical results. We then applied a selection mask defined as a rectangle of $M_{x}\times M_{y}$. The first value of the curve contained in the mask was selected as the vertical position of the active contour in that pixel column. The values $\rho_{1}$, $\rho_{2}$ and γ were empirically set to 8, 260, and 10 pixels, respectively, for optimal performance on the whole dataset. As can be seen in Figure 8k, the resulting boundary accurately follows the bottom boundary of the ONL layer except for the columns with the shading produced by blood vessels. Therefore, a low-pass filtering with a sliding window of N=101 pixels was applied to the contour. Figure 8l shows the resulting smoothed contour.

As previously mentioned, one of the biggest challenges in the segmentation of the intermediate layers was to isolate the effect of shading due to vessels. To this end, we outlined a mask delimiting the areas of the image where the shading was present. Figure 10 details the process of computing such a vessel mask, $I^{\mathrm{vm}}$.

The first step in the mask construction was a lineal filtering of the image in the frequency domain,
(15)$$I_{1}^{\mathrm{vm}}={\mathrm{FT}}^{-1}\left\{H_{1}\cdot \mathrm{FT}\left\{I\right\}\right\},$$
where the superscript vm stands for blood vessel mask and $H_{1}=\mathrm{FT}\left\{H_{1}\right\}$ represents the transfer function, namely the two-dimensional discrete Fourier transform of the point spread function of the filter $H_{1}$. Figure 11 depicts both the point spread function $H_{1}$ and its transfer function $H_{1}$. This filter has low-pass and all-pass behavior in the horizontal and vertical directions, respectively. This behavior can be noticed observing the two dark horizontal bars located at the vertical frequency $\omega_{y}=0$ with a bright gap between them located at the frequencies $\omega_{x}=\omega_{y}=0$. This leads to the preservation of all parts of the image except those with variation in horizontal direction, that is $0\;<\;|\omega_{x}|$, which is removed from the image [68]. Note that the shadows cast by blood vessels are vertical edges, i.e., areas with nonzero horizontal variation, being therefore removed by the filter. Thus, this filter leads to a reduction of the shading effect of the vessels while maintaining the horizontal edges of the image which delimit the layers of interest, as can be seen in Figure 10b.

Once the effect of the shadowing was minimized in $I_{1}^{\mathrm{vm}}$, we calculated the modulus of the negative pixels of the difference between these images,
(16)$$I_{2}^{\mathrm{vm}}={\left|I-I_{1}^{\mathrm{vm}}\right|}^{-}=\left\{\begin{array}{cc} -(I-I_{1}^{\mathrm{vm}}), & I-I_{1}^{\mathrm{vm}}<0,\\ 0, & 0\leq I-I_{1}^{\mathrm{vm}}, \end{array}\right.$$
where the resulting image $I_{2}^{\mathrm{vm}}$, shown in Figure 10c, reflects the areas shaded by vessels as well as other areas outside the area of interest. The following morphological operators were then applied to further refine the shaded areas.

The bright details smaller than $l_{5}^{h}$ and $l_{51}^{v}$ were eliminated firstly by applying two opening operators and one reconstruction,
(17)$$\begin{array}{cc} I_{3}^{\mathrm{vm}} & =\gamma_{l_{5}^{h}}\left(I_{2}^{\mathrm{vm}}\right),\\ I_{4}^{\mathrm{vm}} & =\gamma_{l_{51}^{v}}\left(R_{I_{2}^{\mathrm{vm}}}\left(I_{3}^{\mathrm{vm}}\right)\right). \end{array}$$

Figure 10e depicts image $I_{4}^{\mathrm{vm}}$, where the shaded areas are clearly marked. Since this mask is only be used in the application of the active contours to the layers between the RNFL and the ONL, the pixels outside of these layers are removed, as seen in image $I_{5}^{\mathrm{vm}}$ shown in Figure 10f.

We then apply an opening with $l_{51}^{v}$ to remove the bright elements smaller than the structuring element,
(18)$$I_{6}^{\mathrm{vm}}=\gamma_{l_{51}^{v}}\left(I_{5}^{\mathrm{vm}}\right).$$ Finally, a thresholding considering only pixels with values higher than 10% of the maximum of $I_{6}^{\mathrm{vm}}$ was applied, which resulted in the binary mask $I^{\mathrm{vm}}$, shown in Figure 10h.

This blood vessel mask $I^{\mathrm{vm}}$ was used in the refinement of the #2-AB. In particular, an active contour was applied with the initialization of the contour of Figure 8l, using as external forces $I_{8}^{\#2}$ where the zones bounded by the mask $I^{\mathrm{vm}}$ were canceled out. This allowed the contour to conform to the lower boundary of the ONL without the adverse effect of the artifacts in the shaded areas, thus obtaining a smooth contour as expected for retinal layers. Figure 10i depicts the resulting boundary #2-AB. The active contour parameters used were α=0.5, β=1, γ=1, and η=1.

### 2.7. Segmentation of Boundary #3-LB

For the outline of line #3-LB, we proceeded from image *I* and from boundaries #1-UB and #2-AB. Since contour #3-LB typically features a higher number of concavities and convexities than contour #1-UB and the layer RNFL presents a reduced contrast to layer IPL, a vertical shape rectification was first applied to image *I*.

This rectification, which was also implemented in the Spectralis software (see Figure 2), was a flattening of the image considering a given boundary as a reference. This boundary was flattened, positioning the rest of the pixels in each column of the image with reference to this boundary. In this work, #2-AB was used as the reference boundary,
(19)$$I_{R}=R\left(I\right),$$
where $I_{R}$ is the rectified OCT shown in Figure 12a and R(·) denotes the image rectification process with respect to the boundary #2-AB. The same shape rectification was also applied to the blood vessel mask calculated above,
(20)$$I_{R}^{\mathrm{vm}}=R\left(I^{\mathrm{vm}}\right),$$
followed by a morphological dilation with a $l_{5}^{h}$ in order to horizontally widen the areas of this mask $I_{R}^{\mathrm{vm}}$,
(21)$$I_{R,d}^{\mathrm{vm}}=\delta_{l_{5}^{h}}\left(I_{R}^{\mathrm{vm}}\right),$$
where $I_{R,d}^{\mathrm{vm}}$ is the resulting mask shown in Figure 12c, which is used in the further application of an active contour to the image.

We then performed a morphological filtering of the rectified image $I_{R}$, first removing the dark details smaller than ${◯}_{3}$ by applying an opening and reconstruction to the image complement,
(22)$$\begin{array}{cc} I_{1}^{\#3} & =f_{N}\left(I_{R}\right),\\ I_{2}^{\#3} & =\gamma_{{◯}_{3}}\left(I_{1}^{\#3}\right),\\ I_{3}^{\#3} & =R_{I_{1}^{\#3}}\left(I_{2}^{\#3}\right),\\ I_{4}^{\#3} & =f_{N}\left(I_{3}^{\#3}\right). \end{array}$$

Figure 12d–g shows the resulting images. Subsequently bright details initially smaller than ${◯}_{3}$ were removed again by the opening and reconstruction operations,
(23)$$\begin{array}{cc} I_{5}^{\#3} & =\gamma_{{◯}_{3}}\left(I_{4}^{\#3}\right),\\ I_{6}^{\#3} & =R_{I_{4}^{\#3}}\left(I_{5}^{\#3}\right), \end{array}$$
where $I_{5}^{\#3}$ and $I_{6}^{\#3}$ are the depicted in Figure 12h,i.

In order to remove the shaded areas in $I_{6}^{\#3}$, we first removed the dark elements whose horizontal width was less than 31 pixels, using ${□}_{3\times 31}$ as structuring element,
(24)$$I_{7}^{\#3}=\varphi_{{□}_{3\times 31}}\left(I_{6}^{\#3}\right),$$
removing then the bright details with a width of less than 11 pixels wide,
(25)$$I_{8}^{\#3}=\gamma_{{□}_{3\times 11}}\left(I_{7}^{\#3}\right).$$

As can be seen in Figure 12k, the RNFL was now significantly contrasted to the IPL, and in addition, the number of artifacts within the RNFL was substantially reduced. Similar to #1-UB, a vertical gradient was applied to $I_{8}^{\#3}$, but this time, the positive values were removed to detect white-to-black transitions in the downward direction as shown in Figure 12l,
(26)$$I_{9}^{\#3}=\nabla_{v}^{-}\left(I_{8}^{\#3}\right)=\left\{\begin{array}{cc} \nabla_{v}\left(I_{8}^{\#3}\right), & \nabla_{v}\left(I_{8}^{\#3}\right)<0,\\ 0, & 0\leq \nabla_{v}\left(I_{8}^{\#3}\right), \end{array}\right.$$
where $\nabla_{v}\left(I_{8}^{\#3}\right)$ represents the gradient of the image $I_{8}^{\#3}$ along the vertical direction.

To select the boundary of interest from the white-to-black transitions in image $I_{9}^{\#3}$, we applied a process similar to the one carried out in Section 2.6 sketched in Figure 9. We defined the intervals, $M_{x}=\left[\varrho +\rho_{1},\varrho +\rho_{2}\right]$, where ϱ is the vertical rectified position of #1-UB in that column, and $M_{y}=\left[-255,\gamma \right]$, where γ is an empirically determined threshold. The first value of the curve contained in the mask $M_{x}\times M_{y}$ was selected as the vertical position of the contour in that column. The values $\rho_{1}$, $\rho_{2}$, and γ were empirically set to 8, 60 and 3, respectively, for the optimal performance on the whole dataset. Note that the pixel intensity levels were negative in $I_{9}^{\#3}$, although for the sake of better understanding, these values were considered positive in the image representation shown in Figure 12l and left part of Figure 13.

As can be seen in Figure 12m, the resulting boundary accurately follows the bottom boundary of the RNFL layer, though some discontinuities are noticeable at the areas with blood vessel shading.

To remove these artifacts, we slightly smoothed the contour with an averaging window of width N=15. The resulting contour, which is depicted in Figure 12n, was next used as an initialization for the application of an active contour. This led to the lower boundary #3-LB of the desired retinal layer, illustrated in Figure 12o. In this active contour, the image $I_{8}^{\#3}$ was taken for the calculation of the external forces, being excluded the zones delimited by the mask $I_{R,d}^{\mathrm{vm}}$. The remaining contour parameters were α=0.5, β=1, γ=1, η=1, and B-spline as shape function.

### 2.8. RNFL Thickness Calculation

As described in Section 1, the thickness of the RNFL and GCL is important for monitoring the progression of ocular diseases such as glaucoma. In order to compare the results provided by the proposed method with the measurements given by the OCT device, the circumpapillary RNFL thickness was averaged by sectors (T, TS, NS, N, NI, and TI), and the global mean value was also calculated (G). The thickness of the RNFL was calculated by simply subtracting the position of contours #1-UB and #3-LB, resulting in a vector *w* with the thickness of the layer at each *x*-coordinate of the image *I*. The correspondence between the *x*-coordinate and the polar coordinate of the analysis circumference is detailed in Table 2 and depicted in Figure 3. This vector *w* allowed us to calculate the average value of the RNFL thickness in all the different sectors of the eye. Figure 14 displays the resulting measurements for the OCT image used in the description of the method.

## 3. Results

In order to evaluate its performance, the proposed approach was applied to the images in the database and compared with two additional methods. First, it was compared with version 6.9.4.0 of the Spectralis OCT software, which offers an inbuilt tool for automatic segmentation as shown in Figure 2. Second, it was compared with a state-of-the-art method based on a deep learning approach, specifically an adaptation of the method described in [69], based on an encoder–decoder convolutional neural network architecture. In addition to this comparison between methods, the results of the proposed approach were further validated by expert ophthalmologists.

For the development of the deep-learning-based segmentation method, the encoder–decoder architecture proposed in [69] for rodent OCTs was used, but, for this work, the model was specifically trained with human peripapillary OCT images to perform this comparison. The new resulting model is called H-DLpNet (where H refers to human, DL to deep learning, p to peripapillary, and Net stands for neural network). This approach was widely applied in other OCT-based state-of-the-art works intended for glaucoma classification (see e.g., [44]). The fully convolutional network (FCN) architecture was composed of three encoder-decoder blocks. The aim of [69] was to detect the boundaries of three retinal layers (ILM, IPL-INL, and IS-OS) in rodent OCT images. It is well-known that human and rodent retinas present significant differences. The main distinctions are that the ganglion cell layer (GCL) in rodents is not visually distinguishable (app. 2 µm) while in humans is around 20–60 µm. In addition, the third innermost layer, the IPL, on the contrary, is relatively thick in the rodent retina (app. 60 µm). To achieve the objective proposed in this paper and to obtain the ILM and GCL boundary layers and, therefore, the RNFL layer in human images, it is necessary to adapt the approach developed in [69] for the human retina characteristics. Following a transductive transfer learning approach, we retrained the deep learning segmentation method with human peripapillary OCT images to create the H-DLpNet model. The retraining was carried out by making use of a private database, coming from the Oftalvist Ophthalmic Clinic, which consists of 249 human B-scans around the ONH of the retina. In particular, 93 samples were diagnosed by expert ophthalmologists as glaucomatous, whereas 156 circumpapillary images were associated with normal eyes. Note that each B-scan, of dimensions M×N=496×768 pixels, was acquired using the Spectralis OCT system, which allows for obtaining an axial resolution of 4–5 µm. The RNFL mask used to retrain the approach was obtained automatically by the Spectralis OCT segmentation. The hyperparameters used to update the segmentation model were the following: stochastic gradient descent (SGD) optimizer using mini-batches of eight samples with a momentum value of 0.9 and a learning rate of 0.001 initially and reduced by one order after every 20 epochs. The retraining was composed of sixty epochs by optimizing a weighted multiclass logistic function. The retrained model was used to predict the database of human OCT images used in this study and never seen before by the approach.

For illustrative purposes, Figure 15 shows a qualitative assessment of the RNFL segmentation provided by the manual delineation of the ophthalmologist, the proposed approach, the H-DLpNet method and the commercial software of the Spectralis OCT device. These three examples of representative OCT images were arbitrarily chosen for this evaluation, namely the right eye of patients 2, 20, and 69 from the database. Note that only the ground truth of these three cases is available since the manual segmentation of the experts is time consuming and cost expensive. As discussed in Section 4, the manual segmentation of the entire database will be carried out as a future work to provide a trustworthy ground truth.

On these three images, the Dice similarity coefficient [70] was calculated to measure the similarity between the manual segmentation and the proposed method, and the manual segmentation and H-DLpNet method. Table 3 shows the results of this coefficient for each of the sectors of the eye as well as for the overall segmented layer. The Dice coefficient compares quantitatively the segmentation along the entire RNFL provided by two methods. The closer the coefficient is to unity, the more similarity there is between the segmented RNFL areas by the methods.

Next, a quantitative comparison of the results generated by the proposed method, the H-DLpNet method, and the Spectralis OCT device was carried out across the entire database. Note that the Spectralis device does not provide the thickness values at each polar coordinate of the scan but only provides the average values of the RNFL thickness in each sector of the eye. Therefore, these average values of the RNFL thickness were used in the comparison between the three methods.

From the mean values of the layer thickness in each sector (TS, T, TI, NS, N, and NI) in each of the images, the mean value and the standard deviation were calculated over all images of the dataset. These values were also calculated for the mean thickness of the whole RNFL (G). These values were calculated for the three methods, being gathered in Table 4 for the statistical values of the RNFL thickness measurements. Note that the dataset includes healthy and glaucoma patients with large differences in the RNFL thickness, causing large standard deviation values.

The mean and standard deviation of the thickness error were also calculated. This parameter was determined as the difference in the mean RNFL thickness in each sector provided by all three methods. Both RNFL thickness and thickness error were measured in µm. Finally, the mean and standard deviation of the relative error, defined as the quotient between the thickness error and the RNFL thickness determined by the proposed approach, were also calculated. Note, that this calculation was performed again for each sector of the eye as well as the whole RNFL. Table 5 shows these results.

In addition, the Dice similarity coefficient [70] was also computed for the segmentation resulting from the proposed method and the H-DLpNet method for all the images in the database. The measurements provided by the Spectralis software were excluded from this comparison since only the mean value at each sector was available. The mean value and standard deviation of the Dice coefficient calculated over all the images in the dataset are shown in Table 6.

Finally, a time analysis was performed on an Intel i7-4710HQ @ 2.50 GHz of 32 GB of RAM without using GPU processing. The average processing time for each image is 1.818 s with a standard deviation of 0.997 s.

## 4. Discussion

### 4.1. Robustness against Parameter Variations and Speckle Noise

An important aspect to consider is the robustness of the method against variations in the parameters, in particular the size of the structuring elements. The morphological filters (mainly openings and closings), proposed for the coarse detection of the RNFL boundaries, focus on the shape and size of the objects and their relative intensity (a bright object on a dark background or vice versa). For instance, a circular structuring element with a radius greater than 3, i.e., ${◯}_{3}$, is required in Equation (Equation 4) to detect the upper boundary #1-UB. Given that ${◯}_{7}$ is used in the proposed method, there is a wide range of values in which this morphological filtering would provide the desired results. For the sake of clarity, Figure 16 shows the results of the steps of Figure 7b,d, corresponding to the Equations (Equation 4) and (Equation 5), using circular structuring elements with radii 5 and 9. As can be seen, the effect of the morphological operation is visible in the image, but the desired result, i.e., the delineation of the upper boundary, is practically similar in all cases. Likewise, we analyzed the steps shown in Figure 8h–k, corresponding to Equations (Equation 12)–(Equation 14), where an opening, a closing, and a thresholding of the positive vertical gradient are performed. Variations are made on the proposed structuring element ${◯}_{3}$, using in this case radii 1 and 7, as shown in Figure 17. The only noticeable effect between the images appears in the shaded areas by the vessels, which is subsequently compensated by the application of active contours with suppression of external forces by means of the vein mask. It is important to note that the application of the proposed method to a different dataset might require a readjustment of the parameters used in the filtering. Since morphological operators filter the shape (and size) according to the structuring element, the expected maximum and minimum width of the retinal layer under study, measured in pixels, influences the setting of these parameters.

Regarding the influence of speckle noise on the results of the segmentation, it is negligible for most OCT images of the database due to their correct acquisition and their limited number of artifacts. Nevertheless, the bypassing of these filters could cause problems in OCT images with a higher level of noise or number of artifacts. The proposed filtering suppresses with morphological openings the typical bright spots of speckle noise in Equations (Equation 4), (Equation 10) and (Equation 23); and suppresses with morphological closings the dark spots in Equations (Equation 7) and (Equation 23). As indicated above, the parameter of this filtering, i.e., the size of the structuring element, is not critical, and there is a wide range of values in which the performance of the filtering is optimum. The mean value of the standard deviation of the speckle noise within the dataset used in this work is estimated using the NOLSE estimator [71], resulting in σ=0.0088, within the range of σ = [0.0030–0.0292]. To evaluate the robustness to speckle noise of the proposed method, we tested the method over the OCT images with synthetically added speckle noise with standard deviation from σ=0.031 to σ=0.316 (variance from $\sigma^{2}=0.001$ to $\sigma^{2}=0.1$). Figure 18 illustrates the results on an image with a speckle noise estimation of σ=0.0062. The performance of the method is appropriate with noise level below σ<0.316. As can be seen in Figure 18d, by exceeding this level, the #3-LB was not correctly detected, resulting in an incorrect adjustment to the CGL + IPL layer. It should be noted that given the visual degradation achieved in OCT images, this value should be outside of any realistic practical situation.

### 4.2. Analysis of the Performance of the Methods

By first applying a qualitative analysis to the results depicted in Figure 15, we can clearly see that the three methods handle the delineation of the upper boundary of the RNFL with adequate reliability. However, it can be noticed that the H-DLpNet method tends to generate excessive smoothing in bumpy areas, with peaks and valleys, causing certain inaccuracies. By contrast, both the proposed method and Spectralis tend to produce a result closer to the manual segmentation as defined by the experts.

The analysis of the accuracy of the lower boundary of the RNFL is, on the other hand, notoriously difficult. As described in Section 2.1, the speckle noise and the artifacts due to the shadows cast by blood vessels generate inaccuracies in all the segmentation algorithms. However, this lack of information in the shaded areas is also a challenge in the manual segmentation carried out by the experts. This makes it unclear which method shows a higher precision in certain images of the database.

Analyzing the manual segmentations in Figure 15, it can be observed that the H-DLpNet method tends to project upward the layer delineation inside the areas shaded by the vessels, while Spectralist tends to project the layer in the downward direction. The proposed method, on the other hand, tends to be a compromised solution by maintaining a greater continuity of the layer in these areas. In manual segmentation, experts maintain or lower the location of the lower edge in the shaded areas, as roughly performed by the proposed method and Spectralis, respectively. Note that although only three OCT images are shown in these figures, similar results can be observed in most of the images in the database. This inaccuracy in the shaded areas, even for the experts, complicates the quantitative analysis whose metrics are detailed in Section 3.

In addition, it is important to determine the highest accuracy in layer thickness measurement allowed by the resolution of the images. Although the proposed method has subpixel precision, the thickness measurement by the neural network in the image is performed in pixels. Considering a minimum error of one pixel in the delineation of each boundary, two boundaries of the RNFL to be delineated, and a vertical resolution of 3.87 µm/pixel, therefore a difference of less than 7.74 µm between the layer thicknesses provided by either method, should not be significant due to the image resolution.

Table 4 shows the mean value of the RNFL thickness in each section of the eye calculated by the three methods. It can be seen that the mean values of the layers are between a minimum of 67.3 µm and a maximum of 136.2 µm, which correspond to 8.7 and 17.6 times the minimum resolution allowed by the image. Therefore, the layer thicknesses measured by each method are very close to each other, considering the limitation of the image resolution. Note that the proposed method tends to provide slightly greater thicknesses than the other two methods in most sectors.

Focusing on Table 5 with the thickness errors, it can be observed that the smallest and the largest mean values correspond to 6.9 and 17.6 µm, respectively, which again, are acceptable values given the resolution provided by the image. Excluding the overall thickness (G), we can notice that the temporal sector T shows the lowest error of all sectors, both in the mean value and in the standard deviation. This is mainly due to the fact that the presence of blood vessels in this segment is reduced, and therefore, the three methods provide similar measurements. By contrast, sectors such as the temporal superior (TS), temporal inferior (TI), and nasal inferior (NI), with a considerable number of artifacts and shadows, show much greater differences. One sector of particular interest is the nasal sector (N). In the comparison between Spectralis and the proposed method, it can be seen that the mean error is quite high in both eyes, about 17 µm. As can be seen in Figure 15, although some vessels appear in this sector, the mean error is relatively high, due in part to the fact that Spectralis tends to lower the upper edge. Thus, Spectralis provides a lower mean thickness than the proposed method, with the error being even higher than in sectors with more artifacts.

The two right-hand columns of Table 5 show the relative error of the proposed method with the other two methods. It can be seen that analogously to the absolute error, the lowest value corresponds to the overall mean (G) with 0.07 for right eye, and the worst case corresponds to the nasal sector (N) with 0.19 also for the right eye. It is also noteworthy that the error between the proposed method and the H-DLpNet method has a smaller magnitude than with the results provided by Spectralis. The similarity between the results of the proposed method and the H-DLpNet method can also be measured with the Dice similarity coefficient detailed in Table 4. As can be seen, the concordance between the segmented layer is high, since in all sectors, the Dice coefficient is higher than 0.893 in all cases.

We emphasize that both the proposed method and the H-DLpNet method use the same set of images provided by the Spectralis device. However, we cannot be sure whether the Spectralis software uses only the information from the OCT images or makes use of other data available to the device software but not available from the image set. The details of this inbuilt software are not available and, generally, the processing is specific for each particular equipment. This fact makes it difficult to compare measurements between equipment from different brands as well as with implementations based on recent research. For example, as addressed in [25], the authors highlight the differences in manual and automated segmentation in drusen volume from a Heidelberg spectral domain (SD-) and a Zeiss swept-source (SS) PlexElite optical coherence tomography (OCT).

At this point, it is important to remark that the main purpose of the segmentation of the RNFL in peripapillary B-scan OCTs is the assessment of the glaucoma status or progression. To this end, the main objective is the automatic measurement of the evolution of the RNFL thickness over time and thus determines its relationship with the progression of the disease. For this reason, although the results provided by these three methods differ slightly, what is really important for its application in glaucoma screening is the consistency of the chosen method over time. Since the image characteristics producing these small differences between methods are constant over time, the methods studied, including the proposed approach, can be expected to provide this consistency over time. Nevertheless, this hypothesis needs to be verified as this method is used in research on the progression in glaucoma patients.

An important distinguishing feature between the proposed method and the H-DLpNet method is the ability to apply constraints in the segmentation process. In the H-DLpNet method, a training is first performed with a set of labeled images, and then the convolutional neural network is applied to automatically predict the segmentation in the target images. By contrast, the proposed method can directly operate on the images in the database, being able to impose certain restrictions in the segmentation process, such as grey levels of the layers, minimum and maximum thicknesses, periodicity, smoothness or maximum curvature, and layer continuity, among others. The application of these constraints is more challenging in neural-network-based methods and allows for greater robustness in layer detection. In particular, the H-DLpNet method has problems in the segmentation of seven images of the database, which are instead correctly segmented with the proposed method.

Although the proposed method has proven to be robust to image artifacts produced by vessels shadowing and speckle noise of usual practical scenarios (variance less than 0.1), some issues may hinder the correct segmentation of the RNFL. Neither the proposed method nor the H-DLpNet method do not provide satisfactory results in four images of the dataset, in which the retinal layers suffer from the lack of continuity at the edges of the image or where part of the RNFL is cut off in the image area, as exemplified in Figure 19. It should be noted, however, that in clinical practice, the expert repeats the acquisition of such images, as they are considered invalid.

In addition, the approaches to segment the retinal layers in OCT images based on machine learning usually require a huge amount of labeled data, which are difficult to obtain, even more so if the data are derived from manual segmentations. Furthermore, the contours provided by expert ophthalmologists possess certain variability due the subjective knowledge which changes between observers. Therefore, an automatic method for the segmentation of the retinal layers based on “classical” image processing is useful since it is usually faster, it can be used as a standalone technique, or it can provide a large amount of label data to be used in deep learning methods under human supervision.

Finally, the computational load between methods was also compared. The proposed method uses an average processing time of 1.818 s per image, which is substantially less than the average time of 5.169 s used by the H-DLpNet method for the segmentation of each image. Note that as indicated above, the latter method additionally requires a network training process prior to the prediction process. Thus, the proposed method shows a higher efficiency in the segmentation process.

## 5. Conclusions

The automatic segmentation of the retinal nerve fiber layer (RNFL) in peripapillary spectral-domain OCT images is of great relevance in the diagnosis of glaucoma, since the thickness of this retinal layer is a useful biomarker given its relationship to the disease status and progression. In this context, this work proposes a new approach to segment and estimate the thickness of the RNFL from 2D peripapillary B-scan OCT images. The proposed method consists of a set of stages applied sequentially to each OCT image. First, a preprocessing was performed by means of morphological operations in order to reduce speckle noise and unwanted artifacts, followed by a vertical gradient peak detection. According to the result of this detection, a linear low-pass filtering was then applied, followed by a refinement through the application of deformable models. This yielded the boundaries of the RNFL, allowing one to easily determine the RNFL thickness at each angle of the peripapillary OCT scanning. The efficiency of the proposed method is based on the optimized implementation of the morphological operators and on the frequency implementation of the active contours. On the other hand, the establishment of constraints in the segmentation process guarantees the robustness of the estimation even in the case of defective or highly noisy images.

The proposed method was compared with version 6.9.4.0 of the inbuilt software of the Spectralis OCT imaging device as well as with the H-DLpNet method. In addition, the segmentation results were evaluated and validated by ophthalmologists. The three methods analyzed show similar results, except in certain areas of the retina due to artifacts and shadows cast by the blood vessels of the eye. In these areas, the precise positions of the layer boundaries are uncertain even in the manual segmentation conducted by ophthalmologists. Nevertheless, since the key point in determining glaucoma progression is the variation in RNFL thickness over time, the differences in thickness measured by the methods are not significant, as they are consistent over time for each patient. Thus, we can affirm that the proposed method, like the previous ones, is perfectly valid for the study of the evolution of the RNFL thickness. Furthermore, we highlight that the proposed method is computationally more efficient compared to the H-DLpNet method. Together with the greater robustness of the method and the accuracy of the segmentation results, we can state that the proposed method is valid and efficient for the study of the evolution of the RNFL thickness in peripapillary OCT for the assessment of glaucoma.

Being aware of the limited existence of publicly accessible databases with peripapillary OCT images including patient metadata together with prediagnoses of this disease, in order to facilitate future comparisons and validations with the proposed method, the authors will try to publish the dataset used in this work to make it publicly accessible.

Although the proposed method is focused on obtaining the RNFL thickness in peripapillary OCT, the further extension to the case of OCT volumes is a line of future work. In this new context, although it is not critical, the size of the structuring elements should be analyzed to study their relative size in comparison with the retinal layers to be segmented. However, the most important change would occur in the implementation of active contours. In the case of peripapillary OCTs, a closed contour was used since the circular B-scan provides the boundaries of the RNFL as periodic signals. In the case of an OCT volume, the boundaries of the RNFL cannot be considered as periodic signals, and the implementation of the active contours in the frequency domain is not straightforward. It would be necessary to use hidden extensions [51] to satisfy the boundary conditions imposed when using the discrete Fourier transform. The current implementation of the proposed method has been tested on the nonperipapillary OCT volume dataset [20], producing acceptable results in the outermost slices of this volume, where there is enough edge-based information for localizing most of the boundaries. Unlike methods for the 3D segmentation of intraretinal layers based on regional image textures such as, e.g., [21], the proposed method fails in the central slices of the OCT volume since these edge requirements do not occur. At this point, to overcome this situation, the proposed method could be extended to the 3D case to achieve a robust and more efficient implementation than processing 2D slices of a 3D volume, where the deformable model would be a 3D open active mesh with some a priori knowledge incorporated into the deformable model [51,52] to deal with lack of boundaries of the fiber layers.

Future work on the segmentation process will focus on a more accurate positioning of the RNFL boundaries in the shaded areas. For this, it will be necessary to study the effect of shading and deformation caused by blood vessels in order to determine the real position of the RNFL boundaries in these areas. This information can then be applied as constraints in the segmentation process, enhancing its accuracy. Moreover, the efficiency of the process can be improved by optimizing the most computationally expensive steps such as the active contour adaptation in images with the highest number of artifacts. In addition to the technical aspects of the RNFL thickness estimation process, another important line of work will be the application of this method in its main purpose, namely the assessment of glaucoma disease and its evolution, from the data provided by peripapillary OCT images of patients.

## Figures and Tables

**Figure 1 sensors-21-08027-f001:**
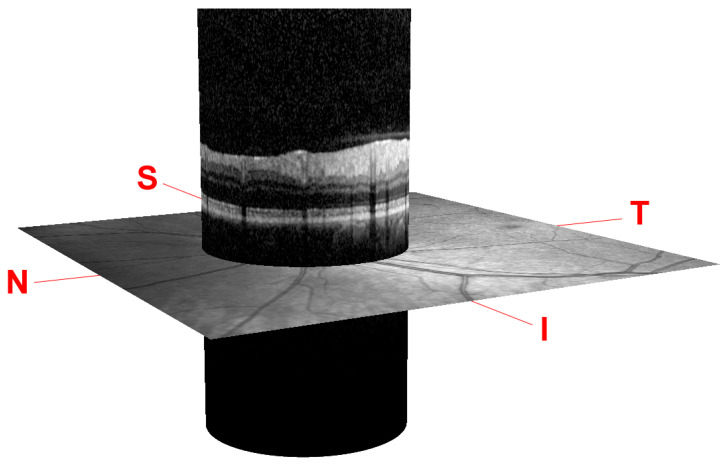
Peripapillary B-scan OCT centered on the optic disc and its relative axial position with respect to a retinal fundus image. Screenshot taken from the Spectralis OCT device. I, S, N, and T stand for inferior, superior, nasal, and temporal, respectively.

**Figure 2 sensors-21-08027-f002:**
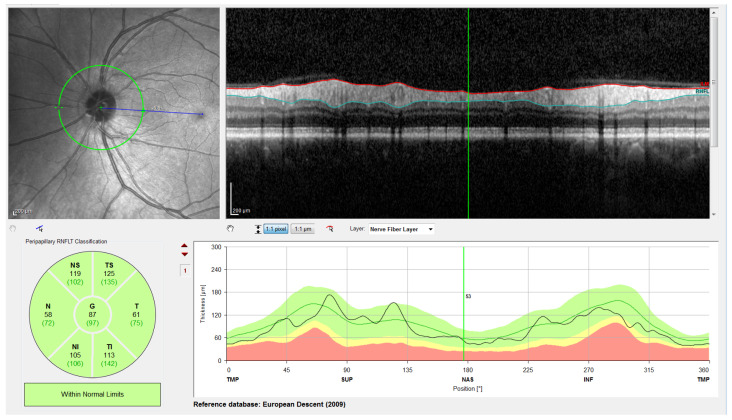
Screenshot provided by the Spectralis software version 6.9.4.0. From left to right and top to bottom: Retinal fundus photography centered on the optic disc (the yellow circle indicates the location of the peripapillary B-scan, which is shown on the right with the segmentation of the RNFL); 2D peripapillary B-scan OCT on Cartesian coordinates; estimated mean values for RNFL layer thickness for the temporal (T), temporal superior (TS), nasal superior (NS), nasal (N), nasal inferior (NI), and temporal inferior (TI) sectors, as well as the overall mean (G); rectified outline of the RNFL with estimated thickness and reference values according to the database European Descent (2009).

**Figure 3 sensors-21-08027-f003:**
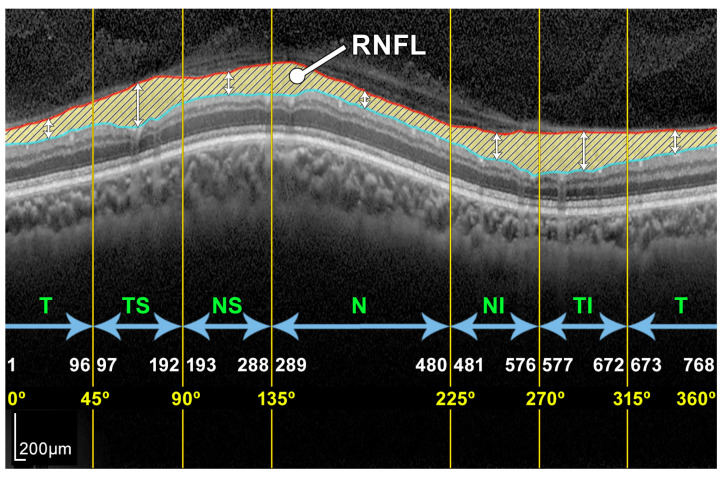
Example of a peripapillary B-scan OCT image with the manual segmentation of the layer of interest RNFL and description of the sectors: T, TS, NS, N, NI, and TI.

**Figure 4 sensors-21-08027-f004:**
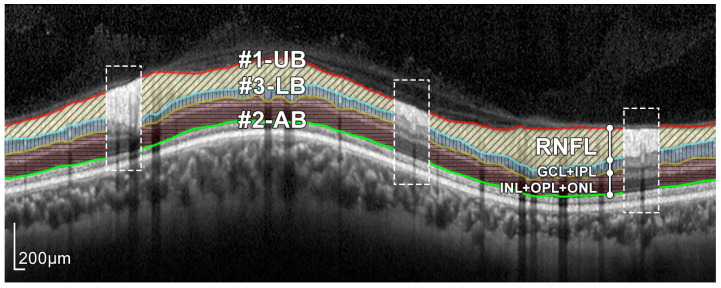
Layers of the retina from top to bottom. RNFL: retinal nerve fiber layer; GCL: ganglion cell layer; IPL: inner plexiform layer; INL: inner nuclear layer; OPL: outer plexiform layer; and ONL: outer nuclear layer. Boundary delineation in the segmentation process. #1-UB: upper boundary of the RNFL; #2-AB: auxiliary boundary corresponding to the lower boundary of the ONL; #3-LB, lower boundary of the RNFL. The dashed boxes exemplify some of the artifacts of the layers.

**Figure 5 sensors-21-08027-f005:**
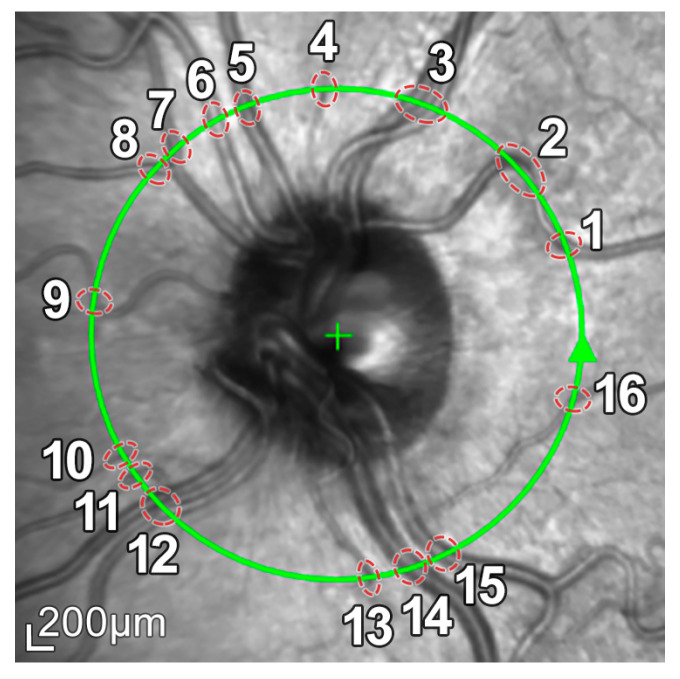
Retinal fundus imaging of the eye under analysis. The green circumference represents the OCT analysis circumference centered on the optic nerve, marked with a **+**. The numbers represent the vessels crossing this analysis circumference.

**Figure 6 sensors-21-08027-f006:**
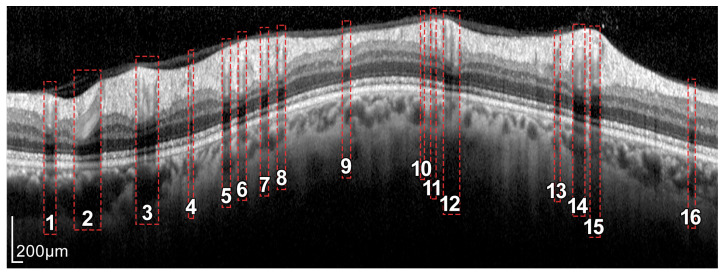
Intensity inhomogeneities and boundary artifacts caused by shadows cast by blood vessels. The numbers identify each of the vessels crossing the circular ray tracing depicted in Figure 5.

**Figure 7 sensors-21-08027-f007:**
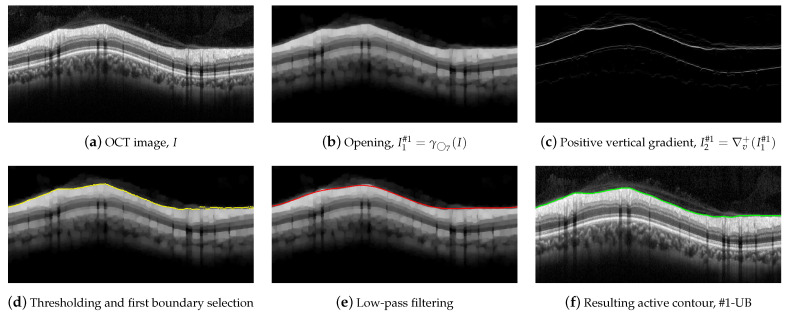
Steps for the delineation of the #1-UB.

**Figure 8 sensors-21-08027-f008:**
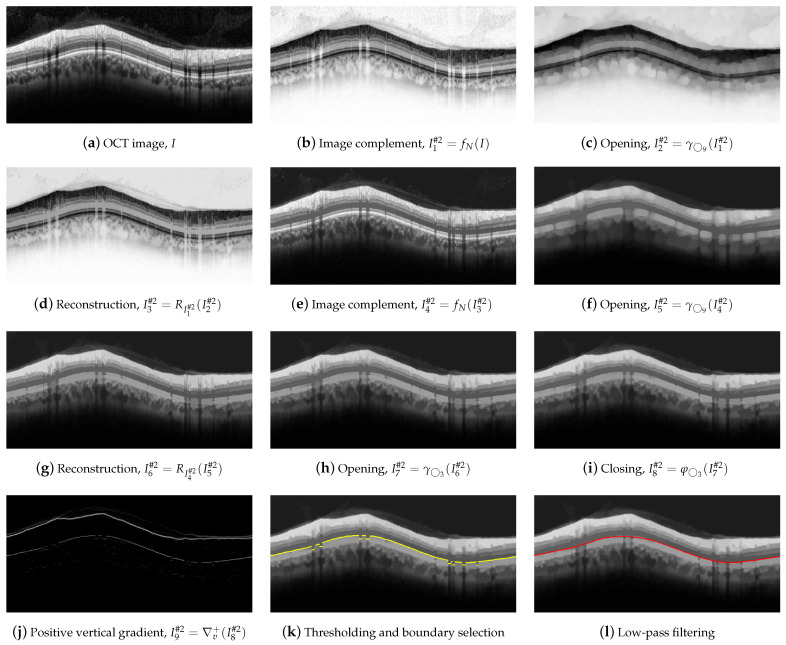
First set of operations applied for #2-AB delineation.

**Figure 9 sensors-21-08027-f009:**
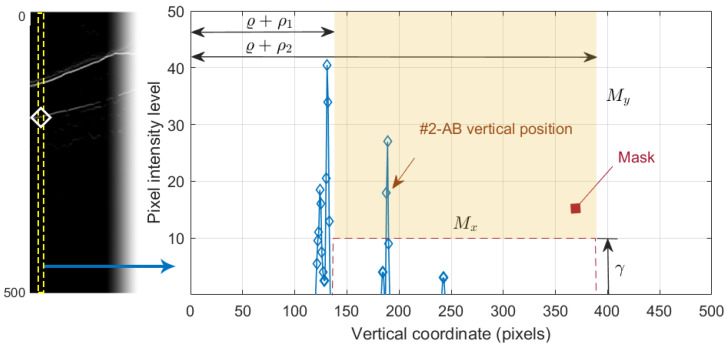
Selection of the #2-AB vertical position. On the left, the positive vertical gradient image $I_{9}^{\#2}$. On the right, 1D profile of the dashed column of $I_{9}^{\#2}$.

**Figure 10 sensors-21-08027-f010:**
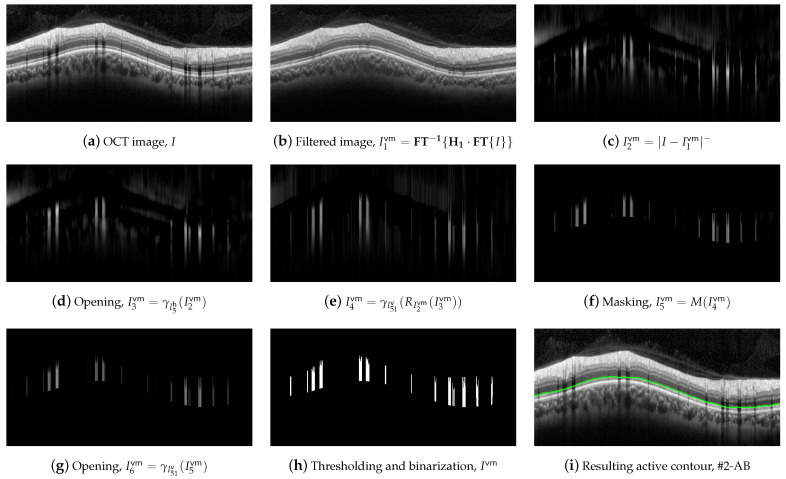
Computation of the blood vessel mask.

**Figure 11 sensors-21-08027-f011:**
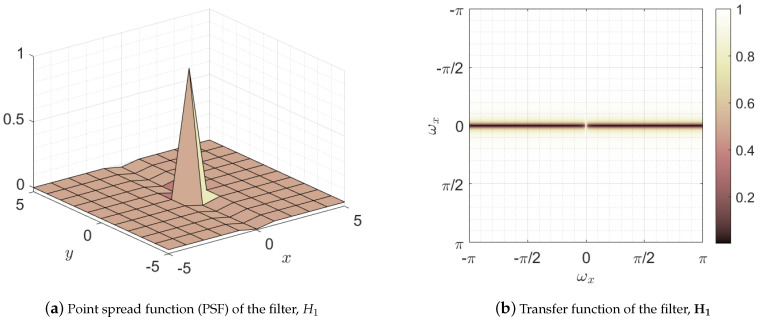
Representation of the filter used in the first stage of blood vessel suppression: (**a**) space domain representation of the filter or point spread function (PSF), and (**b**) frequency domain representation or transfer function.

**Figure 12 sensors-21-08027-f012:**
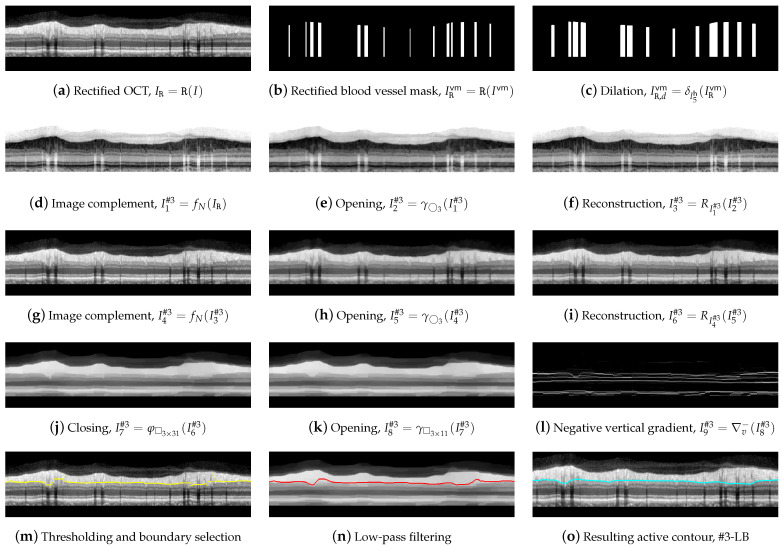
Delineation of the #3-LB.

**Figure 13 sensors-21-08027-f013:**
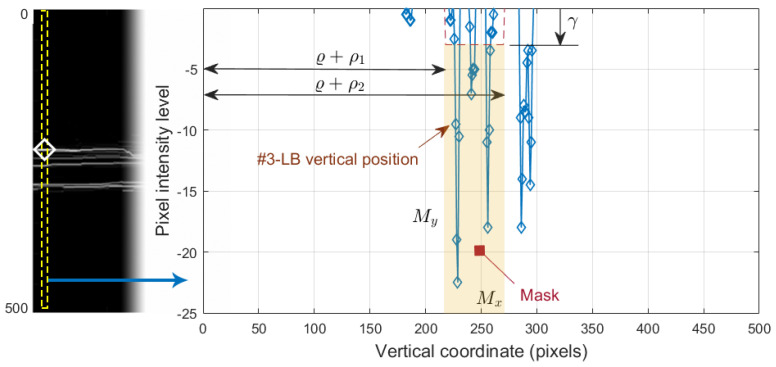
Selection of the #3-LB vertical position. On the left, the negative vertical gradient image $I_{9}^{\#3}$. On the right, 1D profile of the dashed column of $I_{9}^{\#3}$.

**Figure 14 sensors-21-08027-f014:**
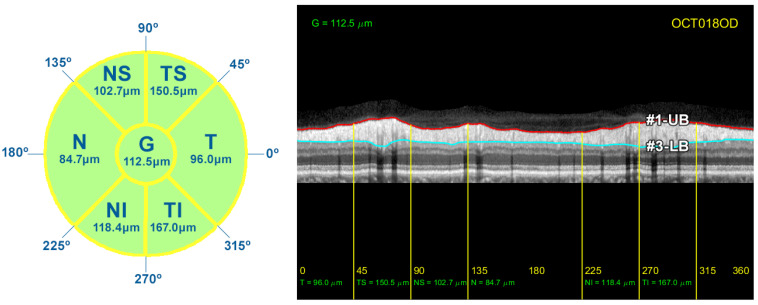
RNFL thickness measurements for each sector of the peripapillary circumference. On the left, diagram detailing the value of the average thickness of the RNFL in each sector of the eye. On the right side, representation of the borders #1-UB and #3-LB, and the average RNFL thickness in these sectors.

**Figure 15 sensors-21-08027-f015:**
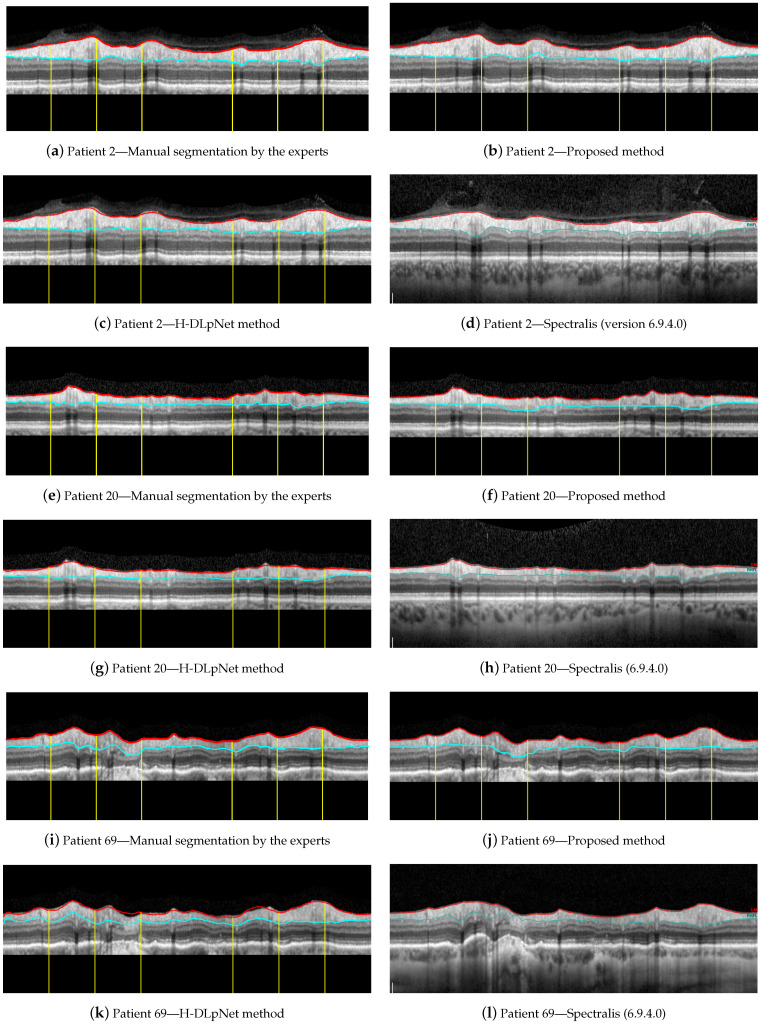
Segmentation results for the right eye of (**a**–**d**) patient 2, (**e**–**h**) patient 20, and (**i**–**l**) patient 69.

**Figure 16 sensors-21-08027-f016:**
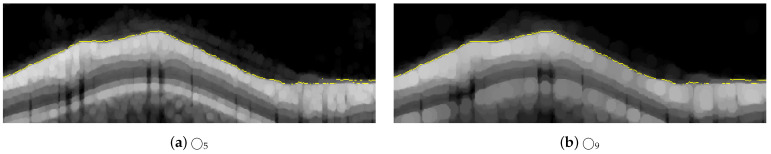
Results of changing the radius of the structuring element from ${◯}_{7}$ to ${◯}_{5}$ and ${◯}_{9}$, used in the steps illustrated in Figure 7b–d for the delineation of #1-UB.

**Figure 17 sensors-21-08027-f017:**
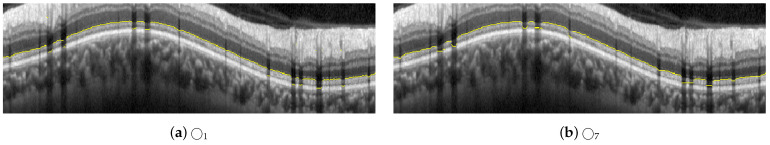
Results of changing the radius of the structuring element from ${◯}_{3}$ to ${◯}_{1}$ and ${◯}_{7}$, used in the steps illustrated in Figure 8h–k for the delineation of #2-AB.

**Figure 18 sensors-21-08027-f018:**
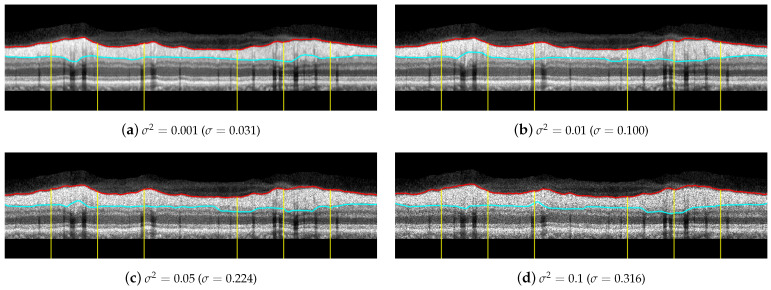
Results of testing the robustness against synthetically incorporated speckle noise from $\sigma^{2}=0.001$ to $\sigma^{2}=0.1$ (σ=0.031 to σ=0.316). The NOLSE estimation [71] of the speckle noise for the raw image is $\sigma^{2}=0.0000381$ (σ=0.0062).

**Figure 19 sensors-21-08027-f019:**
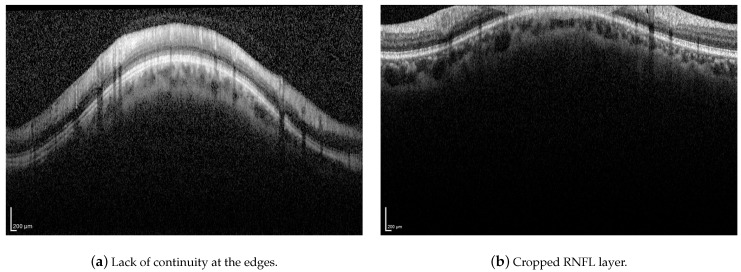
Examples of challenging images for the application of the proposed method.

**Table 1 sensors-21-08027-t001:** Content of the OCT dataset used to validate the approach.

Eye	Healthy	Suspicious	Unhealthy
Left	88	31	43
Right	97	32	38

**Table 2 sensors-21-08027-t002:** Correspondence between the polar and Cartesian coordinates in the OCT projection.

Sector	Polar (Degrees)		Cartesian (Pixels)	
	Min	Max	Min	Max
Temporal (T)	0°	45°	1	96
Temporal Superior (TS)	45°	90°	97	192
Nasal Superior (NS)	90°	135°	193	288
Nasal (N)	135°	225°	289	480
Nasal Inferior (NI)	225°	270°	481	576
Temporal Inferior (TI)	270°	315°	577	672
Temporal (T)	315°	360°	673	768

**Table 3 sensors-21-08027-t003:** Dice similarity coefficient calculated comparing the manual segmentation of the expert with results of the proposed method and H-DLpNet method for each sector of the eye and for the whole layer, G, for the right eye of patients 2, 20, and 69.

Dice Similarity Coefficient						
	Patient 2		Patient 20		Patient 69	
**Sectors**	**Proposed**	**H-DLpNet**	**Proposed**	**H-DLpNet**	**Proposed**	**H-DLpNet**
	**Method**		**Method**		**Method**	
TS	0.974	0.951	0.943	0.927	0.899	0.832
T	0.959	0.927	0.929	0.857	0.927	0.887
TI	0.955	0.954	0.941	0.926	0.947	0.950
NS	0.959	0.912	0.818	0.873	0.851	0.820
N	0.934	0.902	0.896	0.893	0.895	0.850
NI	0.965	0.914	0.943	0.926	0.924	0.886
G	0.956	0.927	0.916	0.901	0.908	0.874

**Table 4 sensors-21-08027-t004:** RNFL thickness (mean and standard deviation in µm) provided by the proposed method, the H-DLpNet method and Spectralis for each sector of the eye (TS, T, TI, NS, N, and NI) as well as for the whole layer (G).

Sectors		RNFL Thickness Measurement (µm)		
		Proposed	H-DLpNet	Spectralis
Left eye	TS	122.8 ± 23.6	128.5 ± 27.4	121.7 ± 30.5
	T	79.0 ± 11.6	70.7 ± 13.9	67.3 ± 13.3
	TI	136.2 ± 30.2	132.7 ± 31.8	127.4 ± 34.8
	NS	116.0 ± 21.4	111.4 ± 28.1	105.5 ± 30.2
	N	86.5 ± 15.7	74.7 ± 16.4	68.7 ± 19.1
	NI	114.2 ± 23.6	109.1 ± 27.1	102.2 ± 28.0
	G	102.5 ± 13.2	96.6 ± 17.0	90.9 ± 17.8
Right eye	TS	126.5 ± 24.9	129.8 ± 25.5	125.2 ± 29.5
	T	80.9 ± 14.0	72.7 ± 15.6	68.6 ± 15.6
	TI	133.4 ± 26.0	131.9 ± 29.9	126.0 ± 34.3
	NS	112.0 ± 23.5	106.0 ± 27.6	98.6 ± 27.4
	N	90.9 ± 14.9	79.6 ± 16.3	72.8 ± 17.9
	NI	113.4 ± 23.5	105.9 ± 25.8	102.6 ± 27.3
	G	103.6 ± 13.6	97.3 ± 16.7	92.0 ± 17.8

**Table 5 sensors-21-08027-t005:** Absolute thickness error (mean and standard deviation in µm) calculated as the difference of the RNFL thickness provided by the proposed approach to both H-DLpNet and Spectralis methods. Relative error (mean and standard deviation) computed for each sector as the absolute error divided by the thickness value determined by the proposed method. These values are calculated for each sector of the eye (TS, T, TI, NS, N, and NI) as well as for the whole layer (G).

Sectors		Absolute Thickness Errors (µm)		Relative Thickness Errors	
		H-DLpNet	Spectralis	H-DLpNet	Spectralis
Left eye	TS	13.1 ± 11.8	14.5 ± 12.9	0.12 ± 0.11	0.13 ± 0.12
	T	8.3 ± 6.1	11.8 ± 7.8	0.11 ± 0.09	0.15 ± 0.10
	TI	10.0 ± 18.1	11.9 ± 10.0	0.08 ± 0.08	0.10 ± 0.11
	NS	12.3 ± 10.9	14.9 ± 12.6	0.11 ± 0.10	0.14 ± 0.12
	N	11.7 ± 11.0	17.6 ± 13.3	0.13 ± 0.12	0.20 ± 0.14
	NI	10.3 ± 9.2	13.5 ± 11.4	0.10 ± 0.09	0.13 ± 0.11
	G	6.9 ± 6.1	10.8 ± 8.0	0.07 ± 0.07	0.11 ± 0.09
Right eye	TS	12.1 ± 10.4	12.8 ± 11.2	0.10 ± 0.10	0.11 ± 0.10
	T	8.3 ± 6.0	12.1 ± 7.9	0.11 ± 0.08	0.15 ± 0.10
	TI	12.8 ± 11.0	15.2 ± 12.4	0.11 ± 0.10	0.13 ± 0.12
	NS	10.1 ± 9.5	15.2 ± 13.5	0.10 ± 0.09	0.14 ± 0.12
	N	11.3 ± 9.3	17.3 ± 12.9	0.13 ± 0.10	0.19 ± 0.14
	NI	10.9 ± 9.6	13.0 ± 10.5	0.10 ± 0.09	0.12 ± 0.10
	G	7.4 ± 6.4	10.4 ± 7.3	0.08 ± 0.07	0.11 ± 0.08

**Table 6 sensors-21-08027-t006:** Dice similarity coefficient (mean and standard deviation) calculated comparing the results of the proposed method and the H-DLpNet method for each sector of the eye and for the whole layer.

Sectors		Dice Similarity Coefficient
		H-DLpNet
Left eye	TS	0.895 ± 0.082
	T	0.918 ± 0.071
	TI	0.921 ± 0.057
	NS	0.898 ± 0.059
	N	0.893 ± 0.078
	NI	0.917 ± 0.057
	G	0.908 ± 0.047
Right eye	TS	0.906 ± 0.059
	T	0.919 ± 0.053
	TI	0.914 ± 0.074
	NS	0.907 ± 0.054
	N	0.902 ± 0.063
	NI	0.917 ± 0.062
	G	0.911 ± 0.047

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
