# Peer review of "Automatic Segmentation of the Retinal Nerve Fiber Layer by Means of Mathematical Morphology and Deformable Models in 2D Optical Coherence Tomography Imaging"

_sensors, 2021, doi:10.3390/s21238027_

Round 1

Reviewer 1 Report

This paper proposed a new approach to segment RNFL in OCT images. For boundary segmentation, the method used active contours with morphological operators to segment objects of interest. The proposed method was compared with the deep-learning approach, Spectralis software, and the results about the thickness and overlap metrics are given.  A few images manually segmented by experts were used for visual comparison between the methods, but quantitive measures of the performance for these datasets could have also been provided. The paper is well organized and written, and the proposed method seems plausible and effective based on the provided experimental results. 

- Minor problems: looks a mismatch OCT image in Fig.15(l) with (i)-(k)

Reviewer 2 Report

In this paper, authors have proposed a two-staged approach towards extracting RNFL from OCT scans in order to assess its thickness levels (which is beneficial for monitoring the progression of Glaucoma). The authors have tested their work on a local dataset (with a promise of releasing it publicly in future). 

Assessing glaucomatous progression through RNFL degeneration is not new and there are already many works which have been published on this. Furthermore, there are also many public datasets available on which the related works are tested and benchmarked. Why the authors have not used these public datasets, and compared their method with the state-of-the-art? In the paper, I could only found the comparison with Amor et al. work [59] but that is primarily designed for the rodent OCT imagery and human eye morphology is quite different than rodents. 

I suggest the authors to test their work at least on one public dataset and compare it with state-of-the-art in a fair manner.

Reviewer 3 Report

  1. There are a fair amount of works which provide thickness maps of retinal layers including RNFL which have not been reviewed and discussed in the paper. For example you can google oct retinal thickness map and find the papers and provide a table about the methods and their pros and cons.
  2. In your method many parameters especially for morphological operators are manually set for different steps. Please discuss about the effect of parameters. If we change the dataset is it necessary to change these parameter? Can you modify your method to automatically or semiautomatically detect the required parameters?
  3. The effect of speckle noise should be discussed clearly. It seems that especially morphological operators are sensitive to the level of noise. So, you can synthetically increase the level of speckle of noise and discuss in which level of noise your will be failed. 
  4. You have to clearly explain in which conditions your method would be successful.
  5. Your method is for obtaining the circumpapillary RNFL thickness map. However you can also use it for parallel B-scans. Please also show the results of your method for all B-scans. In this base and according to the first comment please also obtain the RNFL thickness map of your method. If the parallel B-scans is not available for your dataset you can use the dataset in the next comment.
  6. Please also test your method for other publicly available OCT datasets such as provided data in https://www.ncbi.nlm.nih.gov/pmc/articles/PMC8043121/ and compare your results to others.
  7. According to comment 2, please also discuss about the parameters of your method for this new dataset. Which parameters have the most difference in the new dataset?
  8. The idea of using active contour for layer segmentation of OCT images is not new. For example you can google oct retinal segmentation active contour and find several papers. You have to discuss about these papers and explain about the technical contribution of your method comparing to these techniques. It is important that you clearly explain about the novelty of your method.

Round 2

Reviewer 2 Report

The authors have improved the technical quality of the manuscript and I don’t have aby further suggestions 

Author Response

See the pdf file

Reviewer 3 Report

Unfortunately my comments have not been addressed properly. Specially I think you have to be able to apply your method on each B-scan of parallel macular OCT and then by combining each segmented RNFL from all B-scans make RNFL thickness map. In this base, the authors have not been replied to comments 1, 5, 6, 7. Please revise your paper according to requested modifications in these comments.

In addition, other comments have not been properly inserted in the main text and the replies are only presented in the response letter.

Author Response

Please, refer to the pdf file.

Round 3

Reviewer 3 Report

Since, the method will be failed for high level of speckle noise and also it has not been tested against artifacts, I think this statement should be corrected: "Nevertheless, despite the strengths of the proposed method against image arte645 facts and speckle noise..."

Authors should declare that this method has been tested for this dataset and it may fail for other datasets due to the lack of edge information (such as slices 53 to 80 since in Mahmudi's dataset), and suggest some directions for future works such as adding edge-based image information in localizing most of boundaries, etc.

Author Response

Please, see pdf file.
